# Phytochemical Characterization, Antioxidant and Anti-Inflammatory Effects of *Cleome arabica* L. Fruits Extract against Formalin Induced Chronic Inflammation in Female Wistar Rat: Biochemical, Histological, and In Silico Studies

**DOI:** 10.3390/molecules28010026

**Published:** 2022-12-21

**Authors:** Ikram Allagui, Mabrouk Horchani, Nourhene Zammel, Maroua Jalouli, Abdelfatteh Elfeki, Choumous Kallel, Lamjed Mansour, Salah Alwasel, Abdel Halim Harrath, Hichem Ben Jannet, Mohamed Salah Allagui, Kheiria Hcini

**Affiliations:** 1Laboratory of Animal Physiology, Faculty of Sciences of Sfax, University of Sfax, P.O. Box 95, Sfax 3052, Tunisia; 2Laboratory of Biotechnology and Biomonitoring of the Environment and Oasis Ecosystems, Faculty of Sciences of Gafsa, University Campus Sidi Ahmed Zarroug, University of Gafsa, Gafsa 2112, Tunisia; 3Laboratory of Heterocyclic Chemistry, Natural Products and Reactivity (LR11Es39), Medicinal Chemistry and Natural Products, Faculty of Science of Monastir, University of Monastir, Avenue of Environment, Monastir 5000, Tunisia; 4Laboratory of Histo-Embryology and Cytogenetics, Medicine Faculty of Sfax, University of Sfax, Sfax 3029, Tunisia; 5Department of Biology, College of Science, Imam Mohammad Ibn Saud Islamic University (IMSIU), Riyadh 11623, Saudi Arabia; 6Laboratory of Hematology, University of Sfax, CHU Habib Bourguiba, Sfax 3029, Tunisia; 7Department of Zoology, College of Science, King Saud University, Riyadh 11451, Saudi Arabia; 8Biodiversity, Biotechnology and Climate Change Laboratory (LR11ES09), Department of Life Sciences, Faculty of Science of Tunis, University of Tunis El Manar, Tunis 2092, Tunisia; 9Department of Life Sciences, Faculty of Sciences of Gafsa, University Campus Sidi Ahmed Zarroug, University of Gafsa, Gafsa 2112, Tunisia

**Keywords:** *Cleome arabica* L., phenolic compounds, HPLC, antioxidant activity, potential anti-inflammatory, chronic inflammation, COX-2, inflammatory cytokines mediators, molecular docking

## Abstract

In recent decades, the use of herbs and plants has been of great interest, as they have been the sources of natural products, commonly named as bioactive compounds. In specific, the natural compounds from the Capparaceae family which has been proved to have antioxidant, anti-inflammatory, antimicrobial and anti-carcinogenic activities, by several studies. *Cleome arabica* L. (CA) specie is the most used medicinal plants in Tunisia and elsewhere in North African countries for treatment of various diseases including diabetes, rheumatism, inflammation, cancer, and digestive disorders. The current work was undertaken to estimate the total phenolic, flavonoid and condensed tannin contents, to identify and quantify the polyphenolic compounds, and to evaluate the antioxidant and the anti-inflammatory proprieties of CA fruits extract against formalin induced chronic inflammation in Female Wistar rats. In fact, the antioxidant activity was tested by Diphenyl-1-Picrylhydrazyl free radical scavenging (DPPH), Ferric reducing antioxidant power (FRAP) and Nitric Oxide radical (NO·). Anti-inflammatory effect of fruits extract was examined using formalin (2%) induced paw edema in rats. Molecular docking tools were used to investigate the interaction of some compounds from CA fruits extract with the cyclooxygenase-2 (COX-2) target protein. Our results showed that, the total phenolic, flavonoid and tannins contents, which were assessed by the Folin-Ciocalteu, Quercetin, and Catechin methods, respectively, were 230.22 mg gallic acid equivalent/g dry weight (mg GAE/g DW), 55.08 mg quercetin equivalent/g dry weight (QE/g DW) and 15.17 mg catechin equivalents/g dry weight (CatE/g DW), respectively. HPLC analysis revealed the presence of five polyphenolic compounds whose catechin was found to be the most abundant compounds. The antioxidant activity of extract was quantified by DPPH, FRAP and NO· tests and IC_50_ reached the values of 3.346 mg/mL, 2.306 and 0.023 mg/mL, respectively. Cleome fruits ameliorated the histological integrity of the skin and alleviated the disruptions in hematological parameters (WBC, LYM, RBC, and HGB), inflammatory cytokines (IL-1β, IL-6, TNF-α), C-reactive protein, and some oxidative stress markers (TBARS (−49%) and AOPP (−42%) levels, SOD (+33%) and GPx (+75%) activities, and GSH (+49%) content) induced by formalin injection. Moreover, the in-silico investigation had shown that CA fruits extract compounds have a stronger interaction with COX-2 active site, more than the reference drug “indomethacin” (two H-bonds). Our research gives pharmacological backing to the healthcare utilization of Cleome plant in the treatment of inflammatory diseases and oxidative harm.

## 1. Introduction

Chronic inflammatory illnesses are one of the very common lowering factors of life and are among the most common death causes [1]. The inflammatory response is initiated by mast cell activation associated with the release of several vasoactive components such as bradykinin, histamine, prostaglandins and serotonin [2]. The complex inflammatory process was orchestrated by various pro-inflammatory mediators such as interleukin-1 beta (IL-1β), Interleukin-6 (IL-6), and Tumor necrotic factor-alpha (TNF-α) [3]. The persistence of inflammation increased the risk of several diseases such as cancer, rheumatoid arthritis and skin inflammation [4,5].

Several nonsteroidal anti-inflammatory drugs (NSAIDs) exert their effect by inhibiting cyclooxygenases (COXs) which catalyzes the of arachidonic acid (AA) metabolism in prostaglandins (PGs) and thromboxanes (TXs) [6,7]. COX was shown to exhibit three isoforms, those being COX-1, COX-2 and COX-3 [8]. The constitutive enzyme COX-1 is involved in the production of physiological PGs and TXs which control, respectively, the protection of the gastric mucosa, kidney function and platelet aggregation, acting as a “housekeeper” enzyme [7,9,10]. Further, at an infectious or inflammatory site, COX-2 is induced as a response to the release of pro-inflammatory mediators and therefore catalyzes the release of pathological PGs involved in the inflammatory process [11,12]. Despite their efficacy in treating pain and inflammation [13,14], NSAIDs have serious side effects such as gastrointestinal and cardiovascular complications [15].

In recent years, research on medicinal plants has increased worldwide due to their broad pharmaceutical and/or veterinary phytopharmacology applications [16,17]. In fact, Medicinal plants contain many bioactive molecules which are not only used in traditional medicine and as food’s spices but also in pharmaceutical industry [18,19,20]. Among these molecules the phenolic acids, tannins and flavonoids possess important biological properties such as antimicrobial, anti-inflammatory, anticarcinogenic and antioxidant activities [21,22]. Among these plants, *Cleome arabica* L. (CA) has attracted a great deal of interest in the search for antioxidant and anti-inflammatory molecules. This plant belongs to the Capparaceae family, which includes several species (over 200 species), and it is widely cultivated in the tropics and subtropics of the world, especially in North Africa [23].

CA is a spontaneous Saharan plant, known as “Mnitna” due to its foul smell, which is widely scattered in the sandy zones of Tunisia. This genus is a very significant source of bioactive compounds and has numerous healing and conventional applications [24]. Previous studies have proved CA’s phytotherapeutic benefits of, in alleviating several diseases such as rheumatic pain and cancer [25,26]. Recently, the aqueous extracts are extensively studied as a possible anti-inflammatory treatment [27,28].

Therefore, the purpose of our study was undertaken with the aim of identifying and quantifying the phenolic compounds of *Cleome Arabica* L. fruits extract and evaluating their antioxidant activities. Additionally, we investigated the potential anti-inflammatory effects of fruits extract against formalin induced chronic inflammation in Female Wistar rat using biochemical, and histological assays. Furthermore, Molecular docking was used to characterize the structures of ligand protein complexes and to estimate the free binding energies of docked ligands. This phytochemical characterization of CA fruits extract and their antioxidant and anti-inflammatory effects were carried out in order to revalorize this wild plant as a natural source of bioactive molecules, which are effective in the treatment of pain and inflammation instead of synthetic drugs.

## 2. Results and Discussion

### 2.1. Total Phenolic, Flavonoid and Tannins Contents

Aqueous extract of CA fruits was evaluated for their total phenolic, flavonoid and tannins amounts (Table 1). The total phenolic content in the aqueous extract of CA was 230.22 ± 7.08 mg GAE/g DW. The found amount of flavonoids was around 55.08 ± 1.28 mg QE/g DW. In addition, our results proved that Cleome fruits aqueous extract is rich in tannins (15.17 ± 0.47 CatE/g DW). The phytochemical investigation of the aqueous extract of CA revealed important levels of polyphenols, flavonoids and tannins compared with previous reports [29]. Several studies have indicated that these polyphenols are mainly accountable for the antioxidant potential and anti-inflammatory properties of multiple medicinal plants [30,31]. In addition to phenolic and flavonoid, tannins are widespread and important bioactive molecules in plant. Tannins have a potent antioxidant and antibacterial activities [32,33].

### 2.2. HPLC Analysis of Phenolic Compounds

The HPLC analysis was used to determine the composition of the polyphenolic compounds in CA fruits extract. Peak identification was based on their retention times, their UV–Vis and mass spectra together with data from the literature. Five phenolic compounds (Quercetin, Catechin, Kaempferol, Rosmarinic acid and Naringenin) were identified (Table 2). The major content was picked for catechin 5.45 mg/g followed by quercetin 3.45 mg, kaempferol 3.23 mg/g and naringenin 2.84 mg/g. Our results showed that rosomarinic acid was the minor compound (0.12 mg/g) amid the phenolic compounds mentioned. The HPLC analysis has provided more precise information on the chemical nature of the bioactive compounds abundant in the CA fruits. These data were correlated to an earlier study in which *Cleome arabica* L. was rich in phenolic compounds such as kaempferol and quercetin [23,34,35]. Moreover, several studies have indicated that these polyphenols are mainly accountable for the antioxidant potential and anti-inflammatory properties of multiple medicinal plants [30,31].

### 2.3. Antioxidant Activity

The antioxidant capacity of the CA fruits extract was studied with three assays such as free radical scavenging (DPPH), reducing power (FRAP) and nitric oxide radical (NO·). The results are summarized in Table 3. The aqueous extract exhibited an IC_50_ of DPPH (IC_50_ = 3.346 ± 0.12 mg/mL), FRAP (IC_50_ = 2.306 ± 0.05 mg/mL) and NO· (0.023 ± 0.001 mg/mL) than is higher the IC_50_ of vitamin C which is used as a positive control. In this study, CA can be introduced as an important natural antioxidant to scavenge DPPH, NO· and FRAP radicals. The reducing power of CA fruits extract is important to reduce the Fe^3+^/ferricyanide complex to the ferrous form. Therefore, aqueous extract can scavenge free radicals at low concentrations. Our research’s findings of are coherent with the observation by Chen et al. (2013), who mentioned that the highest antioxidant capacity of phenolic compounds is related to the lower bond dissociation energies of OH groups [36]. In addition, Erdogan Orhan et al. (2019) found that catechin exhibited a strong ability to scavenge DPPH radicals [37].

### 2.4. Anti-Inflammatory Effects of Cleome on Edema Size

As indicated in Figure 1, subcutaneous injection of formalin for 7 days significantly increased the size of edema in FOR group compared to control group, which exhibited normal paw size. The treatment with CA fruits extract (FOR+CA) or INDO (FOR+INDO) mainly reduced the size of edema after the 4th day. As represented in Table 4, Cleome fruits extract inhibited the development of inflammation (55%) more effectively than INDO (45%) (*p* < 0.05). The anti-inflammatory effect of CA was significantly greater than that of INDO from the 5th day (*p* < 0.05).The formalin-induced paw edema was extensively used as a model to evaluate the anti-inflammatory effect of plants in chronic inflammation as it closely resembles human arthritis [38]. The reduction in inflammation by the CA fruits extract decreased the edema size more than positive INDO treatment which was used as a drug reference to compare with, in this work [28]. The CA fruits extract inhibition of inflammation was found to be greater than that induced by INDO as a reference drug (55% vs. 45%). Therefore, this could be partly, caused by the decrease of the release of inflammatory mediators by phytochemicals compounds in CA fruits extract. Although, numerous in vivo studies have shown that different Cleome species have inhibited inflammation in carrageenan-induced paw edema [39]. However, the mechanism pathway remains unclear and/or incomplete.

### 2.5. Hematological Findings

The injection of formalin induces a disruption in hematologic parameters (Figure 2). Lymphocytes proportion and WBC in FOR group was significantly (*p* < 0.01) increased once compared to control group. RBC and HGB were significantly reduced in FOR rats in comparison with healthy rats. Disruptions in hemogram parameters were restored by following treatments of rats with CA or INDO. The inflammatory response consists in the increase of blood vessels permeability leading to the migration and activation of PMNs [40]. In this study, disruptions in hematological parameters (WBC and lymphocytes) were highly significant increase in FOR group. WBC is the key element of the immune system which is related to the initiation of inflammation. They were the predominant cells in chronic inflammation [41]. However, RBC and hemoglobin (Hb) hematological markers were decreased. Reduced levels of Hb emphasized the degradation of premature RBCs and decreased erythropoietin. As a result, an anemic condition developed due to erythrocyte deformation [42,43]. Similar to the previous study, these changes in hematological parameters could be explained by the reduced activity of the immune system [44]. The administration of CA resulted in the restoration of hematological parameters to reach normal levels. Thus, this study is in line with the previous one which showed that the administration of CA alleviated anemia in chronic inflammation [45].

### 2.6. C-Reactive Protein (CRP) and Pro-Inflammatory Cytokines Profiling

As demonstrated in Figure 3, formalin injection increased dramatically CRP levels in FOR group when compared to control. However, a statistically significant decrease of this parameter was observed in rats pre-treated with CA (*p* < 0.05) or INDO (*p* < 0.001) in comparison with FOR group. No significant difference was noted between FOR+CA and FOR+INDO groups. Figure 3 showed the serum levels of pro-inflammatory cytokines in experimental rats. Formalin administration resulted in a rise in IL-1β, IL-6 and TNF-α quantities once compared with the control group. The pre-treatment with Cleome fruits extract decreased these cytokines levels as well, and its effect was superior to that of INDO. Chronic inflammation is characterized by the release of cytokines mediators such as IL-1β, IL-6 and TNF-α. Based on our findings, formalin injection resulted in a rise in CRP, IL-1, IL-6 and TNF-α level in FOR group when compared with control rats. TNF-α, as an important inflammatory cytokine, could stimulate neutrophils to liberate proteases and oxygen-free radicals [46]. During the inflammatory process, IL-6 induces the activation of lymphocytes, promotes endothelial cells to produce chemokines and facilitates inflammatory cells infiltration into the injured site [47]. In general, the anti-inflammatory process is associated with a decrease in nuclear factor-κB (NF-κB) and interleukin levels [48]. CRP is considered a very expedient indicator of inflammation in the tissue. The injection of formalin induced a highly significant rise in CRP marker in FOR group. This confirms the cumulative effect of formalin on chronic inflammation. The acute phase of protein synthesis, as CRP is triggered during inflammatory reactions [49]. This protein was produced by local inflammatory cells in damaged tissues, but it is principally produced in the liver [50]. However, CA fruits extract had importantly lowered the levels of CRP, IL-1, IL-6 and TNF-α in rats, suggesting that the Cleome fruits extract plant has a potential anti-inflammatory effect.

### 2.7. Effect of Cleome Fruits Extract on Oxidative Stress Status

Subcutaneous formalin injection not only led to the production of pro-inflammatory cytokines but also to the release of reactive spices of oxygen (ROS). This chronic inflammation was associated with increased lipid peroxidation and protein oxidation in FOR group. As illustrated in Table 5, MDA and AOPP levels showed an increase in this group when compared to control group. After formalin injection, a significant decrease in SOD and GPx activities was noted as well as a reduction in GSH levels in FOR group. Oxidative stress markers, notably MDA and AOPP, were found to be reduced following treatment with Cleome fruits extract (CA) or INDO as compared to FOR group. Pre-treatment with CA fruits extract or INDO reinstated antioxidant activities and levels to near normal levels. Mainly, the alleviation of the oxidative stress with CA fruits extract was clearly more evident than the reference drug. Oxidative stress is defined as overexpression of reactive oxygen species (ROS) which altered the function of various biological systems. This extra physiological situation is considered as an important pathological disorder associated with formalin chronic inflammation phenomena [51]. Indeed, lipid peroxidation is mostly used as an indicator of oxidative tissue injury leading to inflammation and cell necrosis [52,53]. Therefore, the enzymatic antioxidants such as SOD and GPx and the non-enzymatic antioxidant (GSH) declines are usually considered excellent biomarkers which leads to the weakening of the whole antioxidant system [54]. Oxidative stress is strongly linked to acute and chronic inflammation in the animal model [28,55]. However, this situation was reversed through the administration of CA fruits extract to rats. The administration of Cleome fruits extract to rats had decreased oxidative stress products and increased the antioxidant defense system. The reduction in lipid peroxidation and protein oxidation (AOPP) in rats treated with CA fruits extract could be explained by the enzymatic action and/or free radical scavenging activity of its compounds. Additionally, phenolic compounds such as quercetin and catechin found in many foods and vegetables have potent antioxidant effects. Indeed, oxidants caused by endothelial apoptosis are improved by quercetin, so it is more effective than other antioxidant nutrients (beta-carotene, glutathione and vitamins C and E) [56,57]. This property could be the result of two antioxidant pharmacophores inside the quercetin molecule, which is optimally configured for the sweeping of free radicals [58]. Moreover, its ability to chelate iron and other transition metal ions enabling it also to prevent the iron-catalyzed Fenton reaction [59].

### 2.8. Effect of Cleome Fruits Extract on Formalin-Induced Histological Alterations

In order to assess the anti-inflammatory effect of CA fruits extract against chronic inflammation, we carried out a histopathological analysis of paw tissue from all groups was carried out (Figure 4). FOR group showed acute edema associated with mononuclear inflammatory cell infiltration compared to the Control group with no histological injuries. Examined paw sections of rats receiving CA fruits extract revealed a weak inflammatory reaction. Similarly, the reference drug (INDO) reduced the histological damage induced by formalin. Histological examination of rat paw revealed that formalin injection induced an increase in skin thickness owing to edema development. Edema resulted from the increased vasculature permeability with greater vasodilatation [60]. Skin sections indicated dilatation of the blood vessels as a result of inflammatory cell infiltration. Eltom et al. (2021) [61] detected inflammatory cells infiltration, mainly in the dermal layer and the basal part of the epidermis. These histopathological assessments proved the presence of chronic inflammation. Similarly, Kim et al. (2010) showed the presence of necrotic debris, lymphocyte infiltration and hypertrophy in the subcutaneous regions [62]. The effect of formalin injection in skin tissue has been previously studied [63]. The administration of Cleome fruits extracts protected skin tissue against inflammatory signs.

### 2.9. Molecular Docking Studies

The preventive effect of CA fruits extract may be due to the presence of many phenolic compounds such as quercetin, catechin, kaempferol and flavonoid contents (results approved in our previous research). Therefore, the anti-inflammatory proprieties of plants rich in flavonoids could be induced by the inhibition of some key enzymes involved in the inflammation process, such as COX receptors [64]. In this context, COX-2 enzyme is a target for a wide range of many anti-inflammatory drugs as it is responsible for production of PGs inducing inflammation. The current work investigated by molecular docking the interaction of some compounds from CA fruits extract with the binding site of COX-2. The interaction of formalin and the reference drug with COX-2 active site were analyzed to predict their binding mode and to explain their anti-inflammatory activity using AutoDock vina software. The five identified compounds of the extract (catechin, quercetin, kaempferol, rosmarinic acid, and naringenin) subjected to molecular docking fitted well to COX-2 active site inside the pocket and showed good binding energy scores ranged from −6.5 to −9.8 Kcal/Mol compared to the reference drug INDO (−5.4 Kcal/Mol). Table 6 recapitulated the docking results: binding affinities, number of conventional H-bonds and interacting amino acid residues. The best binding energy was found with catechin (−9.8 Kcal/mol) which displayed six H-bonds in addition to some hydrophobic interactions (Figure 5). Catechin was bonded to His75, Ser339, Leu338, Val335, Gln178 and Ser516 via hydrogen bonds. Kaempferol, the 2nd compound displaying an energy of −9.4 Kcal/mol, had established only two H-bonds. Rosmarinic acid despite having a higher energy value than kaempferol, it showed five H-bonds. INDO was bound to the active site via two H-bonds, Arg106 and Tyr341 (Figure 6). Regarding these obtained data, it may be concluded that in term of conventional H-bonds and binding affinity, the interactions of all CA compounds were much more effective than the reference drug. Therefore, docking study demonstrated that catechin which exhibits the optimal binding energy score and the highest number of H-bond could be a potential therapeutic candidate against COX-2.

These findings confirmed previously reported data on the effects of certain phytochemicals compounds belonging to Cleome through in silico and molecular docking [65,66,67]. Indomethacin created two hydrogen bonds only together with lower interaction scores (−5.4). Hence, indomethacin seemed unassociated with rosmarinic acid (−8.0), and quercetin (−6.5). These compounds were further detailed and compared to indomethacin interactions (Table 6). Moreover, Val 335 and Leu 338 were the most involved COX-2 residues. This may justify their important role in the COX-2 binding and inhibition during the inflammatory process [68]. The 3D illustration and the 2D diagram of interactions exhibit that the selected compounds are bound to COX-2 through a network of electrostatic, hydrophobic and hydrogen bonds [69]. In addition, all the compounds evaluated showed negative binding energy, which may explain the Cleome anti-inflammatory effect as described by the biochemical and histological tests.

## 3. Materials and Methods

### 3.1. Plant Material

The aerial part of CA was collected at the fructification phenological stage between March–April 2019 from Sbeitla-Kasserine (35°14′ N, 9°08′ E, Tunisia). Dr Zouhaier Noumi (Faculty of Science of Sfax, Tunisia) performed the plants identification according to the flora of Tunisia (Pottier-Alapatite, 1979). The reference specimen was deposited in the Herbarium of the Laboratory of Animal Physiology, Faculty of Sciences of Sfax. P.O. Box 95, Sfax 3052, Tunisia. The fresh fruits were separated, washed with distilled water, dried at room temperature for 15 days and crushed to obtain a fine powder.

### 3.2. Characterization of the Plant Extract

#### 3.2.1. Preparation of the Plant Extract

The dried samples (1 g) were macerated in 8 ml of distilled water. The aqueous extract prepared and filtered using Whatman filter paper. Final extracts were kept in vials at 4 °C until their corresponding analyses.

#### 3.2.2. Total Phenolic Content (TPC)

A colorimetric method of Folin-Ciocalteu was used to determine the total phenolic content [70]. The reaction mixture contained 100 μL of samples and 500 μL of Folin-Ciocalteu reagent (10%). The mixture reacted for 3 min before 400 μL Na_2_CO_3_ (2%) was added. The absorbance was measured at 750 nm after 30 min of incubation of the mixture in the dark. The amount of TPC is expressed in mg of gallic acid equivalent per g of dry weight (mg GAE/g DW). Analyses were carried out in triplicate.

#### 3.2.3. Total Flavonoid Content

The total flavonoid content was measured spectrophotometrically in the extract [71]. This procedure consists of the formation of the aluminum-flavonoid complex. A total of 125 μL of the sample was mixed with 37.5 μL NaNO_2_ (5%) and incubated for 5 min. Next, 37.5 μL AlCl_3_ (10%) was added to the mixture. After 6 min of incubation, 250 μL NaOH (1M) and 300 μL H_2_O were added to the mixture. The maximum absorbance was measured at 510 nm. The results were expressed as mg of quercetin equivalent per g of dry weight (mg QE/g DW). All measurements were performed in triplicate.

#### 3.2.4. Determination of Condensed Tannins Content

The tannin assay was performed as follows: 50 μL of the extract, 3 mL of vanillin (4% in methanol), and 1.5 mL of H_2_SO_4_ (9 N in methanol) were mixed. After 15 min, the absorbance was measured at λ = 500 nm [72]. Tannin content was calculated from the catechin regression equation. The tannins concentration was expressed as mg catechin equivalent/g dry weight (mg CatE/g of dry weight).

#### 3.2.5. HPLC Analysis

Phenolic compounds in the Cleome fruits extract were identified using HPLC analysis according to Zheng and Wang, 2001 [73]. The method consisting of a vacuum degasser, autosampler, binary pump with a maximum pressure of 400 bar and an Agilent 1260, Agilent Technologies, Germany, equipped with a reversed-phase C18 analytical column of 4.6 × 100 mm and 3.5 μm particle size (Zorbax Eclipse XDB C18). The DAD detector was set to a scanning range of 200–400 nm. The column temperature was maintained at 25°C. The injected sample volume was 2 μL and the flow rate of the mobile phase was 0.4 mL/min. Mobile phase B was Milli-Q water consisting of 0.1% formic acid and mobile phase A was Methanol. The optimized gradient elution was illustrated as follows: 0–5 min, 10–20% A; 5–10 min, 20–30% A; 10–15 min, 30–50% A; 15–20 min, 50–70% A; 20–25 min, 70–90% A; 25–30 min, 90–50% A; 30–35 min, return to initial conditions. Identification analysis was carried outby comparison of their retention time with those obtained from the extract. For the quantitative analysis, a calibration curve was obtained by plotting the peak area against different concentrations for each identified compound at 280 nm. Phenolic compounds contents were expressed in mg of compounds per gram of dry weight (mg/g DW).

#### 3.2.6. DPPH• Free Radical Scavenging Activity

The antioxidant activity of the extract is based on its ability to scavenge 2,2-diphenyl-1-picrylhydrazyl (DPPH•) radical [74]. A dilution series was prepared for the plant extract. The reaction mixture contains 0.5 mL of the extract and 1 mL of DPPH solution (0.4 mM) in methanol. The mixture was incubated in a dark room for 30 min at room temperature. Ascorbic acid was used as a standard. The reduction in the DPPH- free radical was measured using a spectrophotometer at 517 nm. The percentage inhibition (PI) of the radicals was estimated using the following formula:**%I = ((A_0_ − A_S_)/A_0_) ×100**(1)

A_0_: The absorbance of the control

A_s_: The absorbance of the samples

The results were expressed as the inhibitory concentration of the extract needed to decrease DPPH• absorbance by 50% (IC_50_). Concentrations are expressed as mg/mL.

#### 3.2.7. Ferric Reducing Antioxidant Power (FRAP)

The reducing efficacy determines the antioxidant ability of CA fruits extract to reduce the ferric iron in the ferricyanide complex Fe^3+^ to ferrous iron Fe^2+^. This reduction results in a green color whose intensity is proportional to the reducing power [75]. A total of 2.5 mL phosphate buffer (0.2 M, pH 6.6) and 2.5 mL K_3_[Fe(CN)_6_] (1%) were added to different concentrations of sample (0–1 mg/mL). The mixture was incubated for 20 min at 50 °C. Next, 2.5 mL TCA (10%) was added to their action and centrifuged for 10 min at 3000 rpm. Finally, 2.5 mL distilled water and 0.5 mL of FeCl_3_ were added to the 2.5 mL aliquots. The absorbance was read at 700 nm. The tested compound concentration, which provides 50% inhibition (IC_50_, expressed in mg/mL), was calculated from the graph plotting the inhibition percentage against the extract concentration and confirmed by IBM SPSS software.

#### 3.2.8. Nitric Oxide Radical (NO·) Inhibition

The nitric oxide radical scavenging activity of CA fruits extract was determined using the method described by Green et al. (1982) [76]. The concentration of the test extract, which provided 50% inhibition (IC_50_, expressed as mg/mL), was calculated from the graph representing the percentage inhibition as a function of extract concentration. Ascorbic acid was used as a positive control.

### 3.3. In Vivo Study

#### 3.3.1. Animals and Experiment Design

Female Wistar rats weighting 200–220 g were used in the current study. They were housed in the stainless-steel cages under laboratory conditions of 12 h light/ dark cycle and at temperature and humidity-controlled room. Animals were divided into four groups of six rats each. In this study, the chronic inflammation was induced by subcutaneous plantar injection of formalin (2%) for seven days as previously described [77].

-Control group: rats injected with physiological serum 0.1 mL/kg of body weight and served as a control group.-FOR group: rats injected by 0.1 mL of 2% formalin for 7 days.-FOR+CA group: rats received Cleome fruits extract (50 mg/kg of body weight), by gavage, 30 min before formalin injection.-FOR+INDO group: rats received indomethacin (10 mg/kg of body weight), by gavage, 30 min before formalin injection.

Cleome fruits extract (CA) and indomethacin (INDO) were administered to rats one week prior to formalin injection (2%). In the infected model, the increase in paw edema was measured by vernier caliper method [78].

#### 3.3.2. Sample Collection

Blood samples were collected by decapitation and putting the rat’s neck into EDTA tubes for the determination of hematological parameters. Other blood samples were putted in heparin tubes, and they used for serum separation by centrifugation. The obtained serum was stored at −80 °C until analyzed.

To obtain homogenate from the inflamed skin, tissues (edema) were dissected out and diluted with phosphate buffer saline (pH = 7.4) and centrifuged for 20 min at 9000 rpm. The obtained supernatant was collected and used for oxidative stress testing.

#### 3.3.3. Hematological Parameters Determination

Red Blood Cells count (RBC, 10^6^/μL), White Blood Cells count (WBC, 10^3^/μL), Lymphocytes proportion (Lym, %) and Hemoglobin (HGB, g/dL) were quantified using an automatic SYSMEX SERIE KX-21N hematology analyzer, CHU Habib Bourguiba, Sfax, Tunisia.

#### 3.3.4. Determination of Inflammatory Mediators

C-reactive protein (CRP) is a specific producer of inflammatory process. It increases in proportion to its intensity [79]. The reactive protein was measured by the turbidimetric method using an automatic analyzer COBAS INTEGRA 400″ C-Reactive. The CRP is expressed in mg/L.

IL-1β, IL-6, and TNF-α levels were measured in the plasma samples by enzyme-linked immunosorbent assay using commercially ELISA kits (R&D System, Minneapolis, MN, USA), according to the manufacturer’s instructions. The results were expressed as µg/mL.

#### 3.3.5. Assessment of Oxidative Stress Parameters in Paw Tissue

The initial step was to determine protein quantity in skin tissue according to the method of Oh (1951) [80]. The lipid peroxidation was assessed by analyzing the level of malondialdehyde (MDA) following the method designed by Fraga et al. (1988) [81]. The amount of MDA was measured at 532 nm and expressed as nmoL/mg of protein.

Advanced oxidation of protein products (AOPP) was measured spectrophotometrically at 340 nm as previously described by Witko et al. (1996) [82]. The concentration of AOPP was calculated using the extinction coefficient of 261 cm/mM. The results were expressed as µmoL/mg of protein.

The method developed by Beauchamp and Fridovich [83] was used to evaluate superoxide dismutase (SOD) activity. SOD activity was measured at 580 nm and given as U/mg of protein. Glutathione peroxidase (GPx) activity was determined by the procedure of Flohe and Gunzler [84]. Data are expressed as µmoL of GSH/mg of protein. Glutathione reductase (GSH) level was assessed using the method of Jollow et al. (1974) [85]. The obtained results were expressed as µmoL/mg of proteins.

#### 3.3.6. Histopathological Examination

Paraffin portions of dermal tissues were cut into small sections of 7 μm, then stained with hematoxylin-eosin solution (H&E) and examined under light microscopy at 40× magnifications.

### 3.4. Molecular Docking Procedure

Molecular docking simulation was performed by Auto Dock 4.2 program package [86]. The X-ray crystallographic structure of cyclooxygenase-2 (PDB: 3LN1) with selective inhibitor celecoxib was downloaded from the RSCB protein databank [87]. COX-2 enzyme/protein (3LN1) contains the four symmetrical chains A, B, C and D with different native ligands named BOG (β-Octylglucoside), CEL (celecoxib), HEM (Protoporphyrin IX contains Fe) and last NAG (N-acetyl-D-glucosamine). The chain A was selected for the docking studies. The receptor grid was generated (with all default parameters) around the native ligand CEL (celecoxib) by picking it. The optimization of all the geometries of compounds was carried out using ACD (3D viewer) software (http://www.filefacts.com/acd3d-viewer-freeware-info (accessed on 30 January 2010)). The visualization and analysis of interactions were performed using Discovery Studio 2017R2 (https://www.3dsbiovia.com/products/collaborative-science/biovia-discovery-studio/ (accessed on 30 January 2010)).

### 3.5. Statistical Analysis

The data are expressed as mean ± standard error of the mean (SEM). The obtained results were analyzed statistically by ANOVA followed by the Tukey method for post-hoc analysis, using the GraphPad Prism software. P value of less than 0.05 was regarded as statistically significant.

## 4. Conclusions

In this research, Cleome fruits extract has been proven to be an effective potential source of polyphenols and a useful alternative that replaces or even decreases the use of synthetic antioxidants in foods, cosmetics and pharmaceutical products. The efficient correlation between the polyphenolic content and the antioxidant capacity proved that the polyphenolic constituents are responsible for the antioxidant activity of Cleome. Based on our study’s results, it was concluded that formalin sub-chronic administration induces increases in inflammatory cytokines (IL-1β, IL-6, TNF-α), C-reactive protein, and oxidative stress (TBARS, AOPP, SOD, GPx, GSH), which is caused by formalin injection in rats. Moreover, the in vivo and in silico studies showed that the aqueous extract of CA possesses a greater anti-inflammatory effect than the reference drug (INDO.). This potential could be associated to the existence of phytochemicals natural compounds such as flavonoids (quercetin and kaempferol). Therefore, these results could be exploited in the development of new natural pharmaceutical drugs that have anti-edematous, anti-inflammatory and antioxidant activities with fewer side effects.

## Figures and Tables

**Figure 1 molecules-28-00026-f001:**
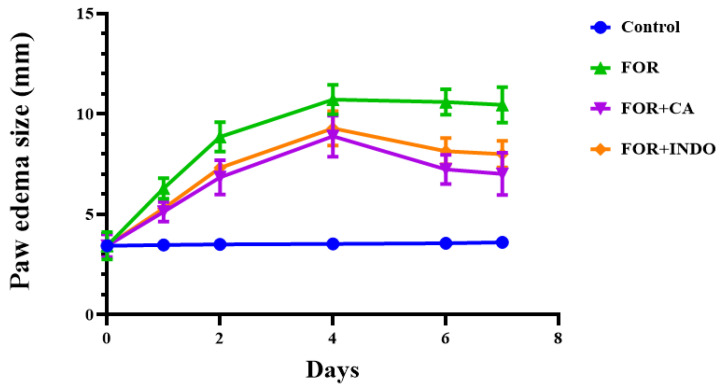
The evolution of paw edema size (mm) in control and treated groups: Control (non-treated rats): received injection with isotonic saline solution (NaCl, 0.9%); FOR: rats inflamed by formalin (2%); FOR+CA and FOR+INDO: rats inflamed by formalin and treated with CA, or indomethacin, respectively.

**Figure 2 molecules-28-00026-f002:**
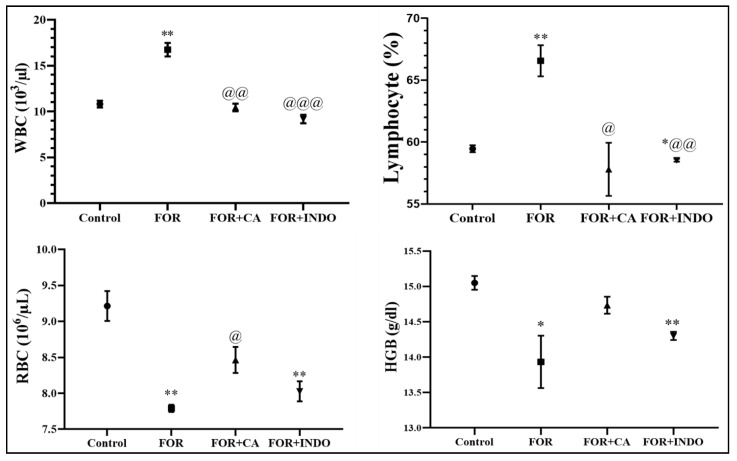
Hemogram parameters of experimental groups. * *p* < 0.05, ** *p* < 0.01 vs. control group; @ *p* < 0.05, @@ *p* <0.01, @@@ *p* < 0.001 vs. FOR group.

**Figure 3 molecules-28-00026-f003:**
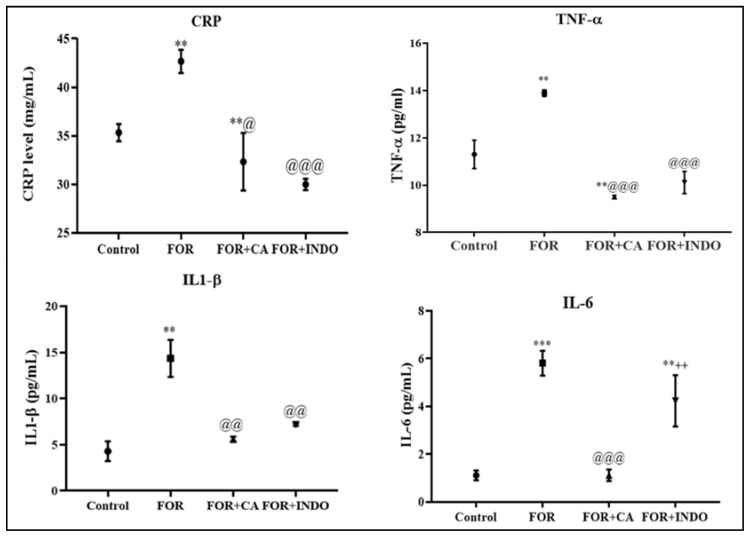
CRP and proinflammatory cytokines levels of experimental groups ** *p* < 0.05, *** *p* < 0.001 vs. control group; @ *p* < 0.05, @@@ *p* < 0.01, @@ *p* < 0.01 vs. FOR group; ++ *p* < 0.01 vs. FOR+CA group.

**Figure 4 molecules-28-00026-f004:**
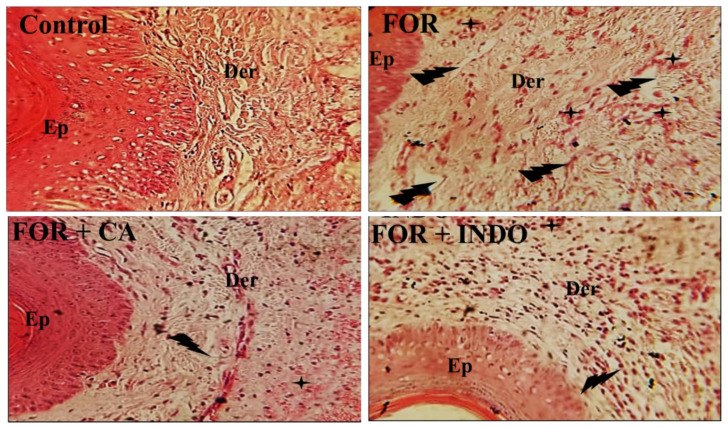
Photomicrographs of paw tissues from the studied groups. The control rats treated with physiological saline (NaCl, 0.9%) showed normal histological structure of both epidermis and dermis; FOR group received formalin 2% showed cumulative edema with massive infiltration of inflammatory cells and paw edema; FOR+CA and FOR+INDO groups treated with CA and INDO, respectively, showing very weak inflammatory reaction once compared with FOR group. Ep: epidermis, Der: dermis: inflammatory cells infiltration; edema.

**Figure 5 molecules-28-00026-f005:**
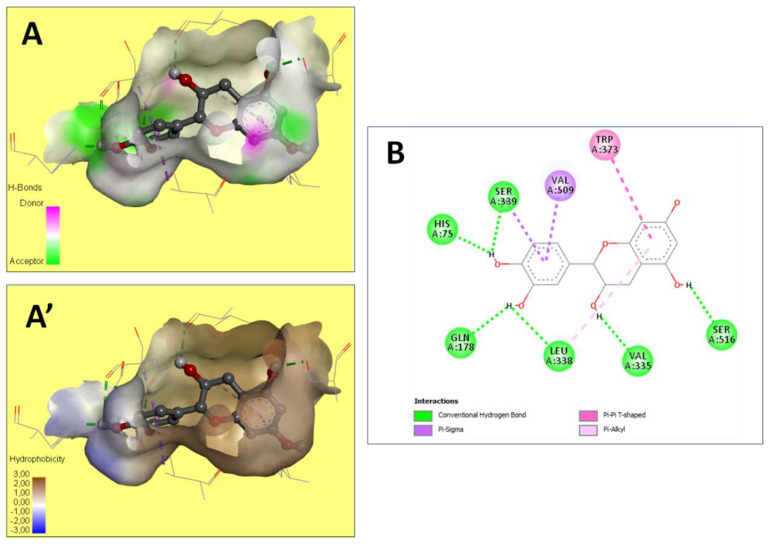
The representation of the 3D structure of «Catechin» with the best docking scores, bounded to the pocket region of COX-2. Micrographs of the pocket region with hydrogen bond (**A**) and hydrophobicity illustrations (**A′**) and the corresponding 2D diagram of interactions (**B**).

**Figure 6 molecules-28-00026-f006:**
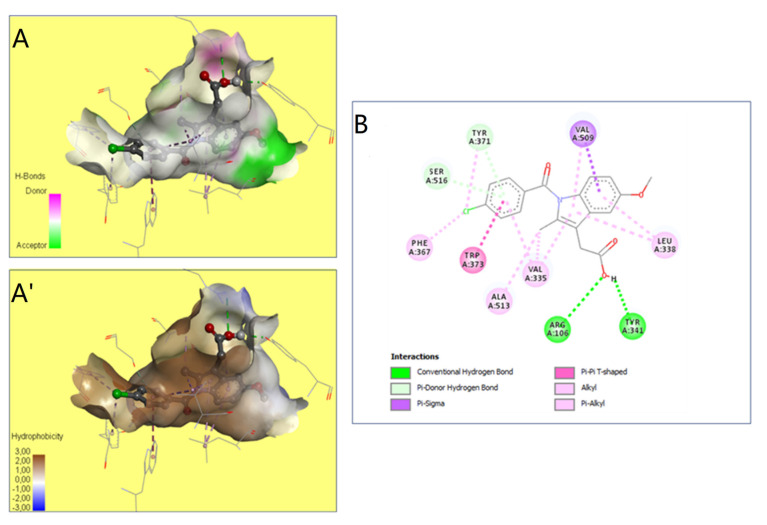
The representation of the 3D structure of «Indomethacin» bounded to the pocket region of COX-2. Micrographs of the pocket region with hydrogen bond (**A**) and hydrophobicity illustrations (**A′**) and the corresponding 2D diagram of interactions (**B**).

**Table 1 molecules-28-00026-t001:** Total phenolic, flavonoid and tannins contents of Cleome fruits extract.

Extract	Total Phenolic Content (mg GAE/g DW)	Total Flavonoids Content (mg QE/g DW)	Total Tannins Content (mg CatE/g DW)
Aqueous	230.22 ± 7.08	55.08 ± 1.28	15.17 ± 0.47

The data represented the means ± ESM for three replicates.

**Table 2 molecules-28-00026-t002:** Phenolic compounds identified in Cleome fruits extract by HPLC.

Phenolic Compounds	RT (min)	Quantification (mg/g DW)	Chemical Structure
Quercetin	5.98	3.45	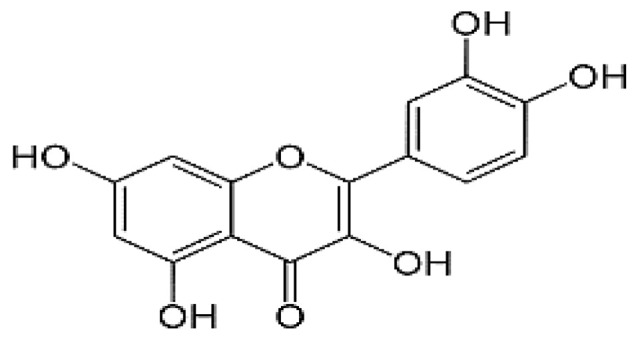
Cathechin	13.78	5.42	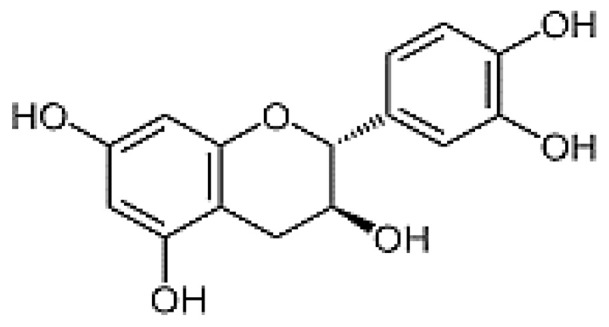
Kaempferol	15.21	3.23	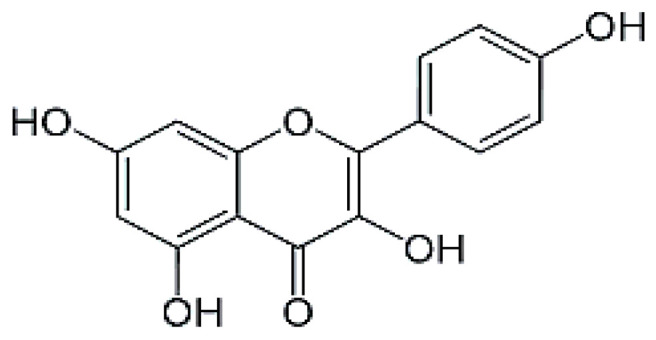
Rosmarinic acid	21.78	0.12	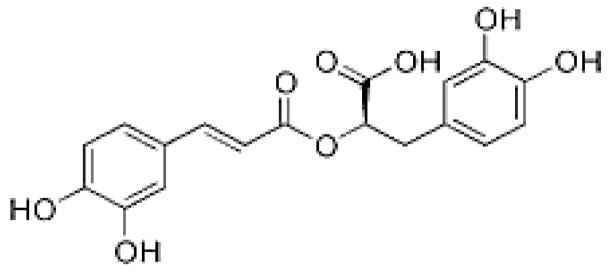
Naringenin	24.45	2.84	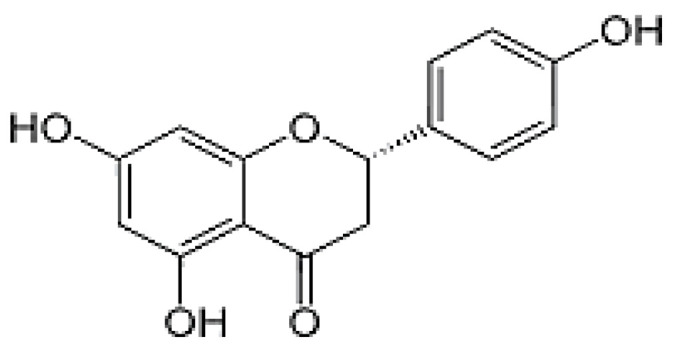

RT: retention time.

**Table 3 molecules-28-00026-t003:** The antioxidant activity of Cleome fruits extract.

Extract	DPPH IC_50_ (mg/mL)	FRAP IC_50_ (mg/mL)	NO· IC_50_ (mg/mL)
Aqueous	3.346 ± 0.12	2.306 ± 0.14	0.023 ± 0.001
Vit C	0.032 ± 0.0005	0.154 ± 0.002	0.006 ± 0.001

The data represented the means ± ESM for three replicates.

**Table 4 molecules-28-00026-t004:** The effects of Cleome fruits extract on paw thickness, % of inflammation and inhibition of paw edema in Formalin-induced chronic inflammation in rats.

Groups	Mean Paw Thickness (mm) at Zero Time	Mean Paw Thickness (mm) after 7 Days	Mean Increase in Paw Thickness (mm) after 7 Days	Inflammation (%)	Inhibition of Inflammation (%)
Control	3.43 ± 0.07	3.5 ± 0.03	0.17 ± 0.01	2.04 ^NS^	-
FOR	3.79 ± 0.28	9.77 ± 0.36 ***	5.98 ± 0.26 ***	157.78 ***	-
FOR+CA	3.67 ± 0.23	6.36 ± 0.43 ***	2.69 ± 0.27 **	73.29 ***	55.01
FOR+INDO	3.32 ± 0.11	6.6 ± 0.12 ***	3.28 ± 0.09 **	98.79 ***	45.15

The data are expressed as mean ± SEM; *n* = 6 animals in each group; *** *p* < 0.001; ** *p* < 0.01 versus Control; NS: non-significant.

**Table 5 molecules-28-00026-t005:** The oxidative stress markers, SOD and GPx antioxidant activities and GSH levels in paw tissue homogenate of control and treated rats.

Groups	Control	FOR	FOR+CA	FOR+INDO
MDA (nmol MDA/g of protein)	11.5 ± 0.63	14.61 ± 0.22 **	9.76 ± 0.18 ^@@@^	11.56 ± 0.07
AOPP (µmol/mg of protein)	1.2 ± 0.03	1.44 ± 0.02 ***	1.01 ± 0.007 **^@@@^	0.97 ± 0.007 ***
SOD (U/mg of protein)	1.9 ± 0.09	1.16 ± 0.006 **	1.74 ± 1.74 ^@@^	1.49 ± 0.003 *
GPx (µmol of GSH/mg of protein/min)	3.29 ± 0.13	0.75 ± 0.04 ***	3.11 ± 0.19 ^@@@^	2.95 ± 0.2
GSH (µmol GSH/mg of protein)	1.19 ± 0.03	0.52 ± 0.002 ***	1.02 ± 0.002 ***^@@@^	1.05 ± 0.008 ***

The data are expressed as mean ± SEM; *n* = 6 animals in each group; *** difference significative compared to control (*p* < 0.001); * *p* < 0.05, ** *p* < 0.01, *** *p* < 0.001 vs. control group; @@ *p* < 0.01, @@@ *p* < 0.001 vs. FOR group.

**Table 6 molecules-28-00026-t006:** The ligands and COX-2 interactions: binding affinity, number of conventional hydrogen bonds and interacting amino acid residues.

Ligand	Binding Affinity (kcal/mol)	Intermolecular Interactions	
		Conventional Hydrogen Bonds	Interacting Amino Acid Residues
Quercetin ^(C)^	−6.5	3	Val102, Arg106 *, Val335, Leu338, Tyr341 *, Leu345, Tyr371 *, Ala513, Leu517
Catechin ^(C)^	−9.8	6	His75 *, Gln178 *, Val335 *, Leu338 *, Ser339 *, Trp373, Val509, Ser516 *
Kaempferol ^(C)^	−9.4	2	Val335, Leu338, Tyr341 *, Tyr371 *, Val509, Ala513,
Rosmarinicacid ^(C)^	−8.0	5	Arg106 *, Leu338 *, Tyr341 *, Phe504, Met508 *, Val509, Gly512, Ser516 *
Naringenin ^(C)^	−7.5	2	Val335, Leu338, Tyr341, Tyr371 *, Val509, Gly512, Ala513, Ser516 *
Indomethacin ^(R)^	−5.4	2	Arg106 *, Val335, Leu338, Tyr341 *, Phe367, Tyr371, Trp373, Val509, Ala513, Ser516

(C): Compound of the extract, (R): Reference, drug, *: Amino acid residue involved in H-bond formation.

## Data Availability

The data presented in this study are available on request from the corresponding author.

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
