# Peer review of "Phytochemical Characterization, Antioxidant and Anti-Inflammatory Effects of Cleome arabica L. Fruits Extract against Formalin Induced Chronic Inflammation in Female Wistar Rat: Biochemical, Histological, and In Silico Studies"

_molecules, 2022, doi:10.3390/molecules28010026_

Round 1
Reviewer 1 Report
Correct the errors in sentences for clarity, for example: “Cleome Arabica L. specie is the 28 most medicinal plants used in Tunisia and elsewhere in North African countries for treatment of 29 various diseases including diabetes, rheumatism, inflammation, cancer, and digestive disorders.”
Page 1, line 36: Molecular docking was developed ??
There are a number of errors in this manuscript, I think it need improvement of English language and grammar corrections.
Table 2. The structure are of different size and skewed. Correct these.
Why figure 1data is without standard deviation ?? While in table you have standard deviation.
Author Response
Response to reviewers and editor comments
Dear Editor and Reviewers
In my name, Dr. Kheiria Hcini, and in the names of all the authors, we would like to thank you very much for the time you have devoted and the effort you have provided to correct this manuscript.
Firstly, we would like to thank you for your kind letter and for reviewers’ constructive comments concerning our article (Manuscript 227333985, International Journal of Food Properties). These comments are all valuable and helpful for improving our article. All the authors have seriously discussed about all these comments. According to the reviewers’ comments, we have tried best to modify our manuscript to meet with the requirements of your journal. All revisions of our manuscript were marked using “Track changes” function and highlighted by using yellow colored text. Also the revision of English language was highlighted by using pink colored text. Point-by-point responses to the reviewers are listed below this letter.
Secondly, if there are any other modifications we could make, we would like very much to modify them and we really appreciate your help. Thank you very much for your help.
Response to comments and Suggestions of Reviewer 1
*Correct the errors in sentences for clarity, for example: “Cleome Arabica L. specie is the 28 most medicinal plants used in Tunisia and elsewhere in North African countries for treatment of 29 various diseases including diabetes, rheumatism, inflammation, cancer, and digestive disorders.”
** The sentences were corrected
*Page 1, line 36: Molecular docking was developed ??
**Molecular docking tools were performed
*There are a number of errors in this manuscript; I think it need improvement of English language and grammar corrections.
** English language and grammar corrections were improved (please find the certificate in attachment file for the Editor).
*Table 2. The structures are of different size and skewed. Correct these.
**The size of chemical structures was corrected.
*Why figure 1data is without standard deviation?? While in table you have standard deviation.
**Standard deviation was added to figure 1.
Reviewer 2 Report
The topic is relevant since to elucidate the molecular biological background of plants used in popular medicine and thereby approve their beneficial effects can be usefull even in pharmaceutical developments of new drugs. In this article the molecular background such as signal transducers of anti-inflammatory effects were maped. Also antioxidant effects were elucidated. According to my opinion (based upon a huge amount of literature) the inflammation and oxidative damage is in the background - or at least increases the likelyhood - of most human diseases, such as cardiovascular diseases, cancer, autoimmune diseases, diabetes, etc. Thus the option to treat this with plant derived drugs are foremost importrant, since it can be a cheap solution to carry out primer prevention, which is the best approach to avoid diseases. In summary, chemopreventive agents have a great potential to reduce the likelyhood of most diseases. The rat paw model is very widely used, because it is a relaible and quantitative model. The technical execution of the experiment well done. The results were presented clearly and the statistical methods were also correct -as far as I know. The resultes were also underpined and explained by the literature data. The conclusion is also correct. Moreover, I have not found weak points of this article.
Maybe the relevance of chemoprevention, tha potential utilizations of Cleome Arabica could be better described. But this could be only a minor supplementation of conclusion and I think it is not necessary, since other literature provide enough information about this - if it is not selfevident.
Author Response
Response to reviewers and editor comments
Dear Editor and Reviewers
In my name, Dr. Kheiria Hcini, and in the names of all the authors, we would like to thank you very much for the time you have devoted and the effort you have provided to correct this manuscript (Manuscript ID: molecules-2005298).
Firstly, we would like to thank you for your kind letter and for reviewers’ constructive comments concerning our article “Phytochemical characterization, Antioxidant and Anti-Inflammatory Effects of Cleome arabica L. fruits extract against formalin induced chronic inflammation in Female Wistar rat: Biochemical, Histological, and In Silico Studies”. These comments are all valuable and helpful for improving our article. All the authors have seriously discussed about all these comments. According to the reviewers’ comments, we have tried best to modify our manuscript to meet with the requirements of your journal. All revisions of our manuscript were marked using “Track changes” function and highlighted by using yellow colored text. Also the revision of English language was highlighted by using pink colored text. Point-by-point responses to the reviewers are listed below this letter.
Secondly, if there are any other modifications we could make, we would like very much to modify them and we really appreciate your help. Thank you very much for your help.
Response to comments and Suggestions of Reviewer 2
The topic is relevant since to elucidate the molecular biological background of plants used in popular medicine and thereby approve their beneficial effects can be useful even in pharmaceutical developments of new drugs. In this article the molecular background such as signal transducers of anti-inflammatory effects were mapped. Also antioxidant effects were elucidated. According to my opinion (based upon a huge amount of literature) the inflammation and oxidative damage is in the background - or at least increases the likelyhood - of most human diseases, such as cardiovascular diseases, cancer, autoimmune diseases, diabetes, etc. Thus the option to treat this with plant derived drugs is foremost important, since it can be a cheap solution to carry out primer prevention, which is the best approach to avoid diseases. In summary, chemopreventive agents have a great potential to reduce the likelyhood of most diseases. The rat paw model is very widely used, because it is a reliable and quantitative model. The technical execution of the experiment well done. The results were presented clearly and the statistical methods were also correct -as far as I know. The results were also underpined and explained by the literature data. The conclusion is also correct. Moreover, I have not found weak points of this article.
* Maybe the relevance of chemoprevention, the potential utilizations of Cleome Arabica could be better described. But this could be only a minor supplementation of conclusion and I think it is not necessary, since other literature provide enough information about this - if it is not self-evident.
** We would like to thank you very much for the time you have devoted and the effort you have provided to correct this manuscript.
Reviewer 3 Report
Dear editor; The attached articled was checked.
The manuscript contain interesting information about Phytochemical characterization, Antioxidant and Anti-Inflammatory Effects of Cleome arabica L. fruits extract against formalin induced chronic inflammation in Female Wistar rat: Biochemical, Histological, and In Silico Studies.
I think that this article is well suits to your journal.
It is generally a good work.
The scientific and presentation level of the manuscript is high.
-I have only a few suggestions about the text The plant names can write short after first writing. Please, real the paper and correct them all.
Methodology is intelligible
References were cross checked.
It would be better if the article is supported with more index articles. Here are my suggestions; Exp: Botanical studies have considerably increased in recent years; Plant studies in different fields have considerably increased in recent
Selvi, S., Polat, R., ÇakılcıoÄŸlu, U., Celep, F., Dirmenci, T., ErtuÄŸ, Z.F., 2022. An ethnobotanical review on medicinal plants of the Lamiaceae family in Turkey. Turkish Journal of Botany, 46(4), 283-332. doi:10.55730/1300-008X.2712
Güler, O, Polat R, Karakose M, CakılcıoÄŸlu U, Akbulut, S., 2021. An ethnoveterinary study on plants used for the treatment of livestock diseases in the province of Giresun (Turkey). South African J. of Botany, 142, 53-62.
ÇakılcıoÄŸlu, U., 2020. An ethnobotanical field study; Traditional foods production and medicinal utilization of Gundelia L. species in Tunceli (Turkey). Indian Journal of Traditional Knowledge 19(4), 714-718.
Author Response
Response to reviewers and editor comments
Dear Editor and Reviewers
In my name, Dr. Kheiria Hcini, and in the names of all the authors, we would like to thank you very much for the time you have devoted and the effort you have provided to correct this manuscript (Manuscript ID: molecules-2005298).
Firstly, we would like to thank you for your kind letter and for reviewers’ constructive comments concerning our article Phytochemical characterization, Antioxidant and Anti-Inflammatory Effects of Cleome arabica L. fruits extract against formalin induced chronic inflammation in Female Wistar rat: Biochemical, Histological, and In Silico Studies. These comments are all valuable and helpful for improving our article. All the authors have seriously discussed about all these comments. According to the reviewers’ comments, we have tried best to modify our manuscript to meet with the requirements of your journal. All revisions of our manuscript were marked using “Track changes” function and highlighted by using yellow colored text. Also the revision of English language was highlighted by using pink colored text. Point-by-point responses to the reviewers are listed below this letter.
Secondly, if there are any other modifications we could make, we would like very much to modify them and we really appreciate your help.
Response to comments and Suggestions of Reviewer 3
*I have only a few suggestions about the text the plant names can write short after first writing. Please, real the paper and correct them all.
** We have written the plant name Cleome Arabica with CA after first writing in all the text.
*References were cross checked.
It would be better if the article is supported with more index articles. Here are my suggestions; Exp: Botanical studies have considerably increased in recent years; Plant studies in different fields have considerably increased in recent
** We have added the suggested references in the section 1. Introduction
Introduction paragraph 3 line 79
-In recent years, research on medicinal plants has increased worldwide due to their broad pharmaceutical and/or veterinary phytopharmacology applications (16 = Selvi et al. 2022, 17 = Güler et al. 2021)
Introduction paragraph 3 line 81
-Medicinal plants contain many bioactive molecules of interest not only for traditional medicine but also for the food and pharmaceutical industry (20 = ÇakılcıoÄŸlu, U., 2020)